# Mixed-species flock sizes and compositions influence flock members' success in three field experiments with novel feeders

Todd M. Freeberg[1,2]*, Colton B. Adams[1,2], Charles A. Price[2], Monica Papeş[2,3]

1 Department of Psychology, University of Tennessee, Knoxville, TN, United States of America,
2 Department of Ecology & Evolutionary Biology, University of Tennessee, Knoxville, TN, United States of America, 3 National Institute for Mathematical & Biological Synthesis, University of Tennessee, Knoxville, TN, United States of America

* tfreeber@utk.edu

**Data Availability Statement:** All relevant data are within the paper and its Supporting Information files.

## Abstract

Mixed-species groups and aggregations are quite common and may provide substantial fitness-related benefits to group members. Individuals may benefit from the overall size of the mixed-species group or from the diversity of species present, or both. Here we exposed mixed-species flocks of songbirds (Carolina chickadees, *Poecile carolinensis*, tufted titmice, *Baeolophus bicolor*, and the satellite species attracted to these two species) to three different novel feeder experiments to assess the influence of mixed-species flock size and composition on ability to solve the feeder tasks. We also assessed the potential role of habitat density and traffic noise on birds' ability to solve these tasks. We found that likelihood of solving a novel feeder task was associated with mixed-species flock size and composition, though the specific social factor involved depended on the particular species and on the novel feeder. We did not find an influence of habitat density or background traffic noise on likelihood of solving novel feeder tasks. Overall, our results reveal the importance of variation in mixed-species group size and diversity on foraging success in these songbirds.

## Introduction

Individuals living in groups typically benefit from being part of those groups by improved ability to avoid predation and find food and other important resources [1,2]. It is becoming increasingly clear, furthermore, that these benefits can be just as great–and perhaps greater–for species regularly occurring in mixed-species groups [3–7]. For example, Diana monkeys (*Cercopithecus diana*) and Campbell's monkeys (*C. campbelli*) shift the height they forage in the canopy when in mixed groups, resulting in both species spending less time being vigilant and more time successfully foraging than when they are in groups of only conspecifics [8]. Similarly, downy woodpeckers (*Picoides pubescens*) spend less time being vigilant and more time foraging in mixed-species groups of songbirds than when they are foraging by themselves [9]. Both Carolina chickadees (*Poecile carolinensis*) and tufted titmice (*Baeolophus bicolor*) were better able to exploit seed from a novel feeder when in more species-diverse mixed-

**Funding:** Funding that supported this work was obtained from the Department of Psychology and the College of Arts & Sciences of the University of Tennessee, Knoxville. The funders had no role in study design, data collection and analysis, decision to publish, or preparation of the manuscript.

**Competing interests:** The authors have declared that no competing interests exist.

species flocks with one another and with white-breasted nuthatches (*Sitta carolinensis*) compared to when in less diverse flocks [10]. Additionally, in captive experimental flocks, chickadees were generally quicker to solve novel feeder tasks when in flocks with larger proportions of titmice [11]. Likewise, though, titmice were found to solve novel feeder tasks more quickly the higher the proportion of titmice in their flocks [11]. Titmice appear to be one of the best species at problem solving in a recent comparative study of vocal learning, cognitive processing abilities, and relative brain size [12]. Given this ability of titmice, perhaps individual titmice and individuals of other species benefit from having a higher proportion of titmice–and attending to their behavior–in these mixed-species flocks.

Benefits of living in mixed-species groups could accrue from the greater "cognitive toolkit" in those groups compared to single-species groups–from the greater pool of competence stemming from the diverse perceptual, attentional, personality, and motivational tendencies of different species [13–17]. Such benefits could also simply be a by-product of the fact that mixed-species groups are often larger than single-species groups, and larger groups are more likely to have one individual solve a problem (e.g., food or predator detection), by skill or by chance, than smaller groups [14]. Increased research on mixed-species groups is needed to understand better the specific factors that influence group- and individual-level success at salient problems faced by group members, including avoiding predation, finding food, moving, and staying cohesive [5].

Here we aimed to assess these social factors (flock composition and size) with the aforementioned mixed-species groups of Carolina chickadees, tufted titmice, and white-breasted nuthatches, in three separate novel feeder experiments. We included in our analyses the larger groupings of species that can occur with these three species (e.g., woodpecker and other species). We also included estimates of two key physical environmental variables–traffic noise and physical habitat structure–in our analyses as these were found to influence anti-predatory behavior in mixed-species groups of these three species [18]. These combinations of variable species aggregations, habitat variability, and novel feeder types allowed us to address the following four broad hypotheses in these experiments:

**H1 –Pool of competence benefit.** Individuals in more diverse mixed-species flocks will be more likely to solve novel feeder tasks compared to individuals in less diverse mixed-species flocks. This finding would corroborate the results of the earlier study with chickadees and titmice [10].

**H2 –Group size benefit**. Individuals in larger flocks (more conspecifics or more total individuals of all species) will be more likely to solve novel feeder tasks compared to individuals in smaller flocks. This finding would corroborate the results of studies using novel feeder tasks in other, including related, songbird species [15,19].

**H3 –Habitat density benefit.** Denser habitat and/or greater proximity to vegetation cover permits decreased vigilance in predator detection and increased foraging success (e.g., ruddy turnstones, *Arenaria interpres*: [20]). Under this hypothesis, individuals in flocks accessing feeders in denser surrounding habitat and greater proximity to cover will be more likely to solve the novel feeder tasks than individuals in more open habitat.

**H4 –Traffic noise distraction.** Anthropogenic noise can serve as a perceptual distractor to individuals, making them less likely to behave in adaptive ways (e.g., anti-predator behavior in hermit crabs, *Coenobita clypeatus*: [21]; and in northern cardinals, *Cardinalis cardinalis*: [22]). Under this hypothesis, individuals in flocks in areas with greater traffic noise should be less likely to solve novel feeder tasks than individuals in quieter areas.

## Methods

### General methods

Our three field experiments were conducted between January–March 2022 (Experiments 1 and 2) and between October 2022 –March 2023 (Experiments 2 and 3). We exposed mixed-species flocks to three novel feeder tests (one feeder for each Experiment) at 36 feeder sites at the University of Tennessee Forest Resources Research & Education Center, 35°59′37″ N, 84° 13′15″ W, located near Oak Ridge, Tennessee. Several of our feeder sites are close to either Oak Ridge Highway (TN-62) or to a construction aggregate gravel company, so there is considerable variation across feeder sites in noise levels. Feeder sites also varied in habitat type and density as the study location has undergone different regimes of forest treatments and plantings over the last several decades.

Each feeder site was separated from the nearest feeder site by at least 375 m to help ensure that we were studying different flocks at each feeder [23], since most of the birds in this study were not individually color-banded. Each feeder site had a platform feeder composed of a wooden board (~ 25 × ~ 40 × ~ 2 cm) mounted on the top of a steel pole (1.8 m tall). When each pole was set in the ground, the wooden board was roughly 1.5 m off the ground. Each feeding station stood < 2 m from a small tree or bush to provide perching and cover for birds using the feeder. For half of the feeding stations, we stocked each feeder with ~100 g of a mix of black oil sunflower seed and safflower seed (hereafter, 'regular' seed mix) every 10–14 days in the weeks prior to, and during, our data collection period. Chickadees, titmice, and nuthatches (as well as a few other species such as red-bellied woodpeckers, *Melanerpes carolinus*, and downy woodpeckers) regularly use these feeding stations as food sources after they discover the presence of seed. For the other half of the feeding stations, we stocked each feeder with ~ 100 g of a mix of black oil sunflower seed, safflower seed, striped seed, peanuts, dried cherries and cranberries, red and white millet, cracked corn, and dried mealworms (hereafter, 'diverse' seed mix), to try to increase the number of species present at these feeder sites.

Early on the morning of data collection, we stocked each feeding station with ~50 g of its particular seed mix (whether the regular or the diverse seed mix). If we later observed at least one chickadee and at least one titmouse at the feeding station, we introduced the particular novel feeder test. We first removed any remaining seed from the feeding station (except for Experiment 3, where we left roughly 50 seeds on the feeding station). We were able to set up the novel feeders quickly (< 60 sec), and such manipulation of stimuli on or near the feeders in our studies has negligible impact on the flock presence or composition in our studies.

During each novel feeder trial, we determined the maximum number of individuals of each species we could detect at one time within a roughly 20 m radius of the feeder. We used this "maximum number detected" metric as our estimate of the real number of individuals of each species present at the feeder since the birds were not individually color-marked and since this metric is highly positively correlated with the real number of birds present at a feeder [23]. Each novel feeder trial lasted 30 min. For chickadees, titmice, and nuthatches separately, we coded the latency for an individual to take the first seed from the novel feeder (at 5-sec intervals). We video-recorded trials at a subset of feeder sites for each of our studies to obtain interobserver reliability statistics [24], which we report below for each experiment.

### Novel feeder experiments

Experiment 1 (January–February 2022) involved a squirrel- and large-bird-resistant transparent tube feeder that was placed in a white plastic bowl on top of the feeding station ("Enchanted Garden Squirrel Resistant Cage Feeder", Menard Inc., Eau Claire, Wisconsin;

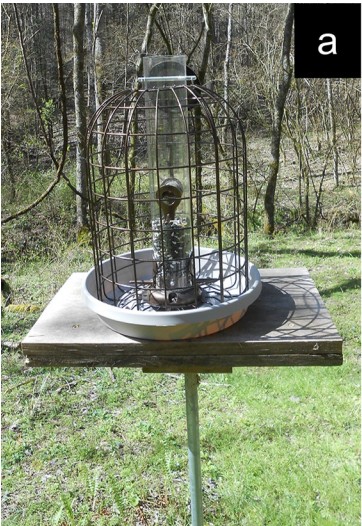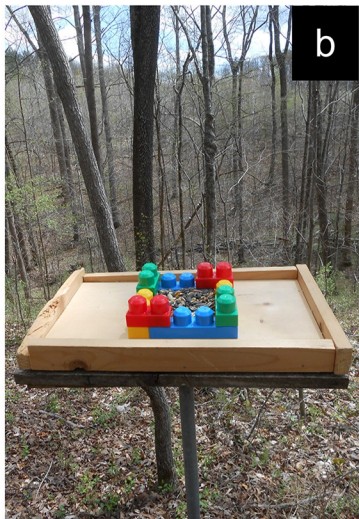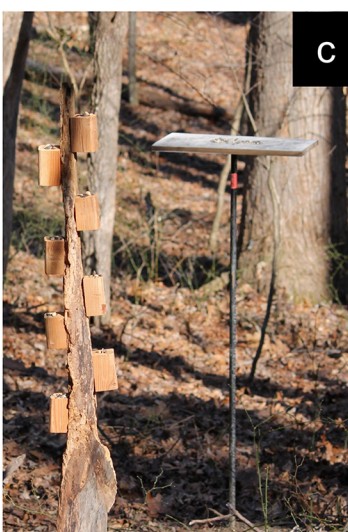

**Fig 1. The three novel feeder contexts assessed.** a = transparent tube feeder with regular seed mix, from Experiment 1. b = plastic blocks feeder with diverse seed mix, from Experiment 2. c = separate food source with diverse seed mix, from Experiment 3. The separate food source structure is 3 m closer to the photographer than the feeding station.

Fig 1A). To obtain a seed from this novel feeder, a bird had to go through the wire grating outside the feeder, inhibit the tendency to peck at the transparent plastic tube to get at the seed, and find one of the two available small hopper trays at the bottom of the tube that contained seed. Although this type of feeder may be commonly used for backyard bird feeding, the flocks of birds at our feeder sites have never experienced a structure like this.

Experiment 2 (February–March and October–December 2022) involved a plastic Mega Bloks construction serving as the seed tray on top of a wooden platform placed on top of the feeding station (MEGA Brands Inc, Montreal, Quebec; Fig 1B). To obtain a seed from this novel feeder, a bird had to stand outside the plastic blocks and reach its beak over to the seed, perch on the construction to obtain a seed, or hop or fly directly into the seed mix to obtain a seed.

Experiment 3 (January–March 2023) involved a separate food source placed 3 m from the original feeding station. This separate food source was created using a 1.5 m long piece of a tree branch, with 8 small wood blocks (~ 4 cm x 4 cm x 7 cm) screwed into the branch, and ~ 2 cm diameter and 2 cm deep cups drilled into the wood blocks for holding seeds (Fig 1C). Each seed cup held 10–15 seeds. The branch was positioned vertically via an attached rebar pole that was buried into the ground. To obtain a seed from this novel feeder, a bird had to fly to it and obtain a seed from one of the seed cups.

## Flock size and composition metrics

We measured three variables related to group size and diversity for chickadees and titmice in all three studies, and for nuthatches in Experiment 3 (the only study where sufficient numbers of nuthatches exploited the novel feeder). **Conspecific flock size** was the number of chickadees, the number of titmice, and the number of nuthatches in the flocks for chickadees, titmice, and nuthatches, respectively. **Mixed-species flock (MSF) size** was the total number of individuals of all species in the flock. A species was counted as being in the flock if at least one individual of that species was present within 10 m of the feeding station for at least 3 min of the 30-min novel feeder trial or if an individual of the species landed on the feeding station

and/or the novel feeder during the trial. The **diversity index** of the flock was assessed as in [10] using the inverse Simpson index [25]. Diversity Index was calculated as $[(P_{chickadees})^2 + (P_{titmice})^2 + (P_{speciesA})^2 + (P_{speciesB})^2 + ... + (P_{speciesZ})^2]^{-1}$, where P is the proportion of chickadees, titmice, and other species present in each flock. For example, the lowest diversity index occurred in Experiment 3 when the flock contained 1 chickadee and 6 titmice (flock diversity index = 1.324) and the highest diversity index also occurred in Experiment 3 when the flock contained 4 chickadees, 2 titmice, 1 white-breasted nuthatch, 1 red-bellied woodpecker (*Melanerpes carolinus*), 1 blue jay (*Cyanocitta cristata*), and 1 brown creeper (*Certhia americana*; flock diversity index = 4.840).

## Environmental variables

We measured anthropogenic noise levels at each feeder site from November 2021 through March 2022. We used a Quest Technologies 2100 Sound Level Meter set at C weighting and slow (1 sec) response to obtain SPL (dB) measures 4–6 times at each site. We measured noise levels between 1000 and 1300 Eastern time on calm days, excluding Sundays. Feeder sites varied considerably in anthropogenic (mainly traffic) noise, with **median noise levels** ranging from ~50 dB to ~70 dB (S1 Fig). Feeder sites also varied in how variable their noise levels were across sampling days.

To assess forest structure near our feeder sites, we used a FARO Focus S 350 HDR scanner to perform terrestrial lidar scans at each of the 36 sites from late January to late March 2022. The composition and density of trees and shrubs varies among our feeder sites, and vegetation type and density may influence anti-predatory behaviors in our mixed-species flock system [18,26]. The lidar scanner was placed on a large tripod such that it was positioned directly above the feeding station. Each scan encompassed a full 360 degrees horizontal and 300 degrees vertical at a scan distance of 50 m, and total scan times were roughly 8 min per feeder. The total number of points collected from lidar scans is a raw metric of vegetation density. We processed all lidar scans (point clouds) in FARO SCENE software (FARO Technologies, Lake Mary, Florida) and saved scans as.las files for additional analyses in CloudCompare (V2 2.13. alpha, Open Source Project), a 3D point cloud and mesh processing software. We cropped each point cloud to remove outliers using statistical outlier removal and noise removal. We also applied a cloth simulation filter to each scan to separate ground from non-ground points (vegetation). Cleaned scans (i.e., those that were filtered and included only vegetation points) were saved as new.las files that were then batch-processed in MATLAB R2023a (MathWorks Inc., Natick, Massachusetts). We used alpha shapes, geometric objects created by lines connecting points that fall within a certain radius of one another, to estimate vegetation volume ($m^3$) and area ($m^2$) from lidar scans [27]. This process creates surfaces and volumes in three dimensions that can be compared across collections of points from each feeder site. We created an alpha shape for each individual point cloud using an alpha radius of 1 m and standard algorithms available within the MATLAB programming environment (alphaShape). The surface area and volume of each resulting **forest alpha shape**–our metric of vegetation volume– were then recorded.

## Statistical analyses and reliability

We tested the effects of our three social and two physical environmental variables for each feeder site–conspecific flock size, MSF size, and diversity index assessed on the day of the novel feeder test, and median noise level and forest alpha shape measured at other times–on birds' success at solving the novel feeder tasks. Following [10], we used binary logistic regression to determine whether obtaining seed from the novel feeder (yes/no) was predicted by one

or more of these variables. To estimate the effect size in each model, we squared the correlation term comparing success or failure at each feeder with the predicted probabilities from the regression model [28]. For the subset of data involving success in each novel feeder experiment, we used multiple linear regression to determine whether one or more of these variables predicted latency to take the first seed from the novel feeder. To estimate the effect size for these latency data in successful flocks, we simply squared the correlation term between latency to solve the task and the particular variable. We analyzed only the full models as our sample size was not large enough for stepwise regression [29]. To assess whether the diverse seed mix attracted more species to the feeders than the regular seed mix, we used two-tailed Mann-Whitney U tests. All statistical analyses were run in IBM SPSS Statistics Version 23. Data on these five social and physical environmental variables, as well as whether flocks were successful and with what latency, are available as supplemental online material (SOM data sets S1–S3 Files).

To obtain data on inter-observer reliability, we video recorded a subset of trials in each of the three experiments using a Panasonic HC-V100M recorder. For the 15 feeders video-recorded in Experiment 1, we agreed on whether chickadees and titmice were successful or not at exploiting seed from the novel feeder 100% of the time, and for successful flocks, our agreement on coding of latencies to take a seed was high (chickadees: $N = 6$, $r = 1.000$; titmice: $N = 9$, $r = 0.996$). For the 13 sites video-recorded in Experiment 2, we agreed on whether birds were successful or not at exploiting seed from the novel feeder 100% and 92.3% of the time for chickadees and titmice, respectively. For successful flocks, our agreement on latencies to take a seed was high (chickadees: $N = 7$, $r = 0.982$; titmice: $N = 8$, $r = 0.993$). Finally, for the 12 sites video-recorded in Experiment 3, we agreed on whether chickadees, titmice, and nuthatches were successful or not at exploiting seed from the novel feeder 100% of the time, and for successful flocks, our agreement on latencies to take a seed was high (chickadees: $N = 5$, $r = 0.993$; titmice: $N = 6$, $r = 0.883$; nuthatches: $N = 4$, $r = 1.000$).

All our methods were carried out in accordance with published guidelines of the Animal Behavior Society, Association for the Study of Animal Behaviour, and the Ornithological Council. The experiments conducted here were approved by the University of Tennessee's Institutional Animal Care and Use Committee (Freeberg protocol 1248).

## Results

### Experiment 1: Transparent tube feeder

The mean ± SD (range) number of chickadees in flocks was 3.42 ± 1.68 (1–8) individuals. There was an average of 4.53 ± 2.18 (1–10) titmice and 1.39 ± 0.77 (0–2) nuthatches in the flocks. Mean MSF size was 11.03 ± 2.80 (5–17) individuals and mean diversity index across flocks was 3.03 ± 0.89 (1.47–4.80). Of the 30 flocks that contained at least one nuthatch, nuthatches took seed at only 2 of the transparent tube feeders. Feeders with the diverse seed mix attracted a median of 5 birds (range 2–6) compared to a median of 3.5 birds (range 2–5) for feeders with the regular seed mix ($n_1 = 18$, $n_2 = 18$, $U = 44$, $p = 0.001$).

Carolina chickadee flocks took seed from the transparent tube feeder at 14 of the 36 sites. The only factor that predicted chickadee flock success was the flock's diversity index ($B = 2.736$, $SE = 0.984$, $Wald = 7.738$, $p = 0.005$, $r^2 = 0.479$). Successful chickadee flocks were in flocks with a higher diversity index than unsuccessful chickadee flocks (Fig 2). The next largest Wald statistic was for the number of chickadees ($B = 0.418$, $SE = 0.348$, $Wald = 1.444$, $p = 0.230$). For successful chickadee flocks, there was a tendency for chickadees to be quicker to take seed the greater the number of conspecifics in their flocks ($B = -7.731$, $SE = 4.126$, $t_1 = -1.874$, $p = 0.098$, $r^2 = 0.310$).

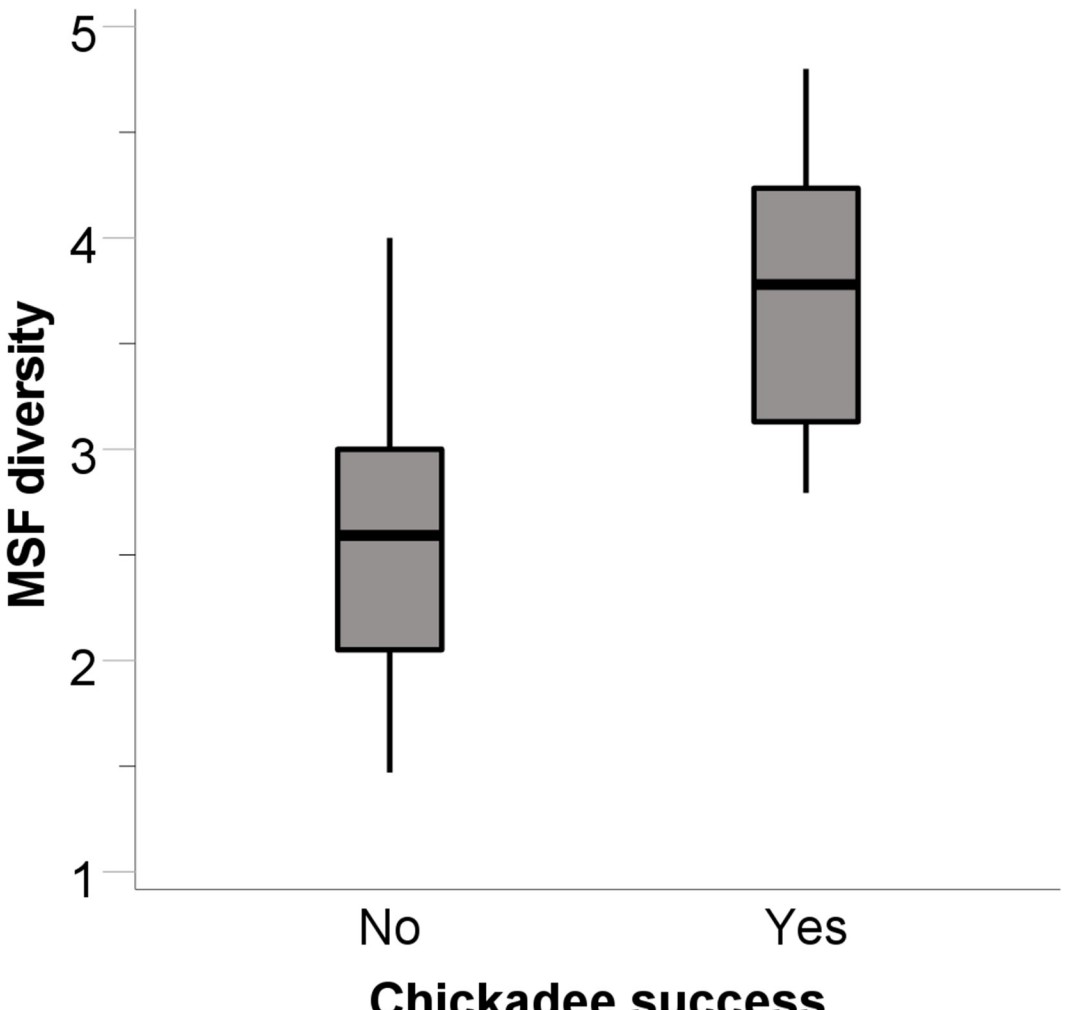

**Fig 2. Carolina chickadee success at solving the transparent tube feeder related to mixed-species flock (MSF) diversity.**
Data are medians (thick black lines), 25- and 75-percentiles (gray boxes), and ranges not including outliers (whiskers).

Tufted titmouse flocks exploited seed from 20 of the 36 transparent tube feeders. The only factor that predicted titmouse flock success was the conspecific flock size (B = 1.400, SE = 0.505, Wald = 7.687, p = 0.006, $r^2$ = 0.536). Successful titmouse flocks were in flocks with more conspecifics than unsuccessful titmouse flocks (Fig 3). The next largest Wald statistic was for flock diversity index (B = 0.387, SE = 0.792, Wald = 0.238, p = 0.625). For successful titmouse flocks, furthermore, there was a tendency for the latency to take seed to be predicted by both conspecific flock size (B = -2.923, SE = 1.449, $t_5$ = -2.017, p = 0.063, $r^2$ = 0.194) and forest alpha volume (B = -0.003, SE = 0.001, $t_5$ = -1.859, p = 0.084, $r^2$ = 0.153). Titmice in successful flocks tended to take seed quicker the greater the number of conspecifics and the less dense the habitat surrounding the feeder.

## Experiment 2: Plastic blocks feeder

The mean ± SD (range) number of chickadees in flocks in this experiment was 3.36 ± 1.44 (1–7) individuals. Flocks contained an average of 4.03 ± 1.38 (1–7) titmice and 1.06 ± 0.96 (0–3)

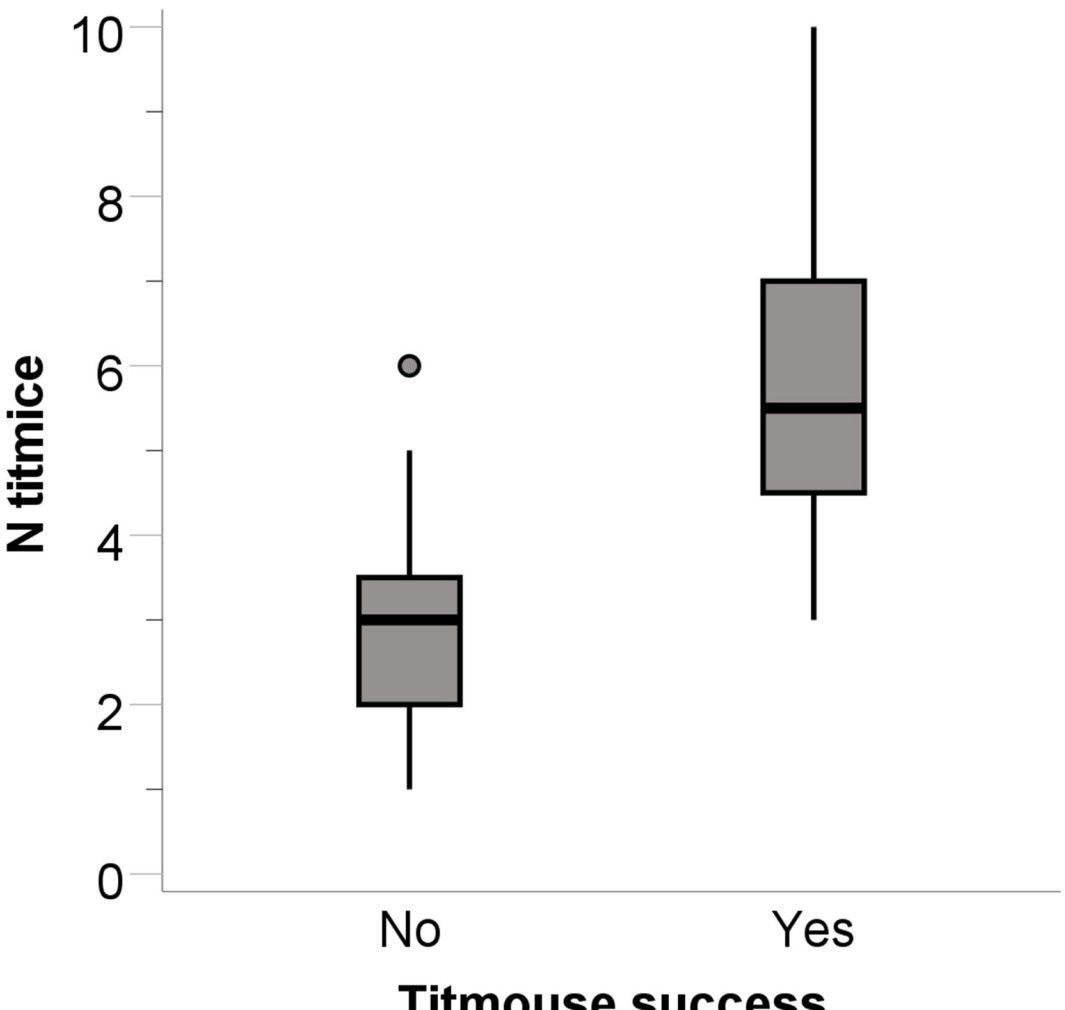

**Fig 3. Tufted titmouse success at solving the transparent tube feeder compared to the number of titmice in the flock.**
Data plotted as for Fig 2, with addition of outlier (circle).

nuthatches. Mean MSF size was 9.64 ± 3.07 (3–17) individuals and mean diversity index across flocks was 2.83 ± 0.78 (1.60–4.67). Feeders with the diverse seed mix attracted a median of 4 birds (range 2–7) compared to a median of 3 birds (range 2–4) for feeders with the regular seed mix ($n_1$ = 18, $n_2$ = 18, U = 69, p = 0.002).

Carolina chickadee flocks took seed from the plastic blocks feeder at 9 of the 36 sites. There was a tendency for the flock diversity index to predict chickadee flock success (B = 3.295, SE = 1.788, Wald = 3.396, p = 0.065, $r^2$ = 0.460)–successful chickadee flocks tended to be in flocks with greater diversity of species than unsuccessful chickadee flocks. For successful chickadee flocks, the latency to take seed was predicted by the conspecific flock size (B = 5.673, SE = 1.165, $t_5$ = 4.869, p = 0.017, $r^2$ = 0.691). Chickadees in successful chickadee flocks were quicker to take seed when there were *fewer* conspecifics (Fig 4).

Tufted titmouse flocks obtained seed from 20 of the 36 plastic blocks feeders. The two factors that predicted titmouse success were the flock diversity index (B = 2.053, SE = 0.949, Wald = 4.677, p = 0.031) and the conspecific flock size (B = 1.121, SE = 0.530, Wald = 4.468, p = 0.035, $r^2$ = 0.295). Successful titmouse flocks were in flocks with greater diversity of species

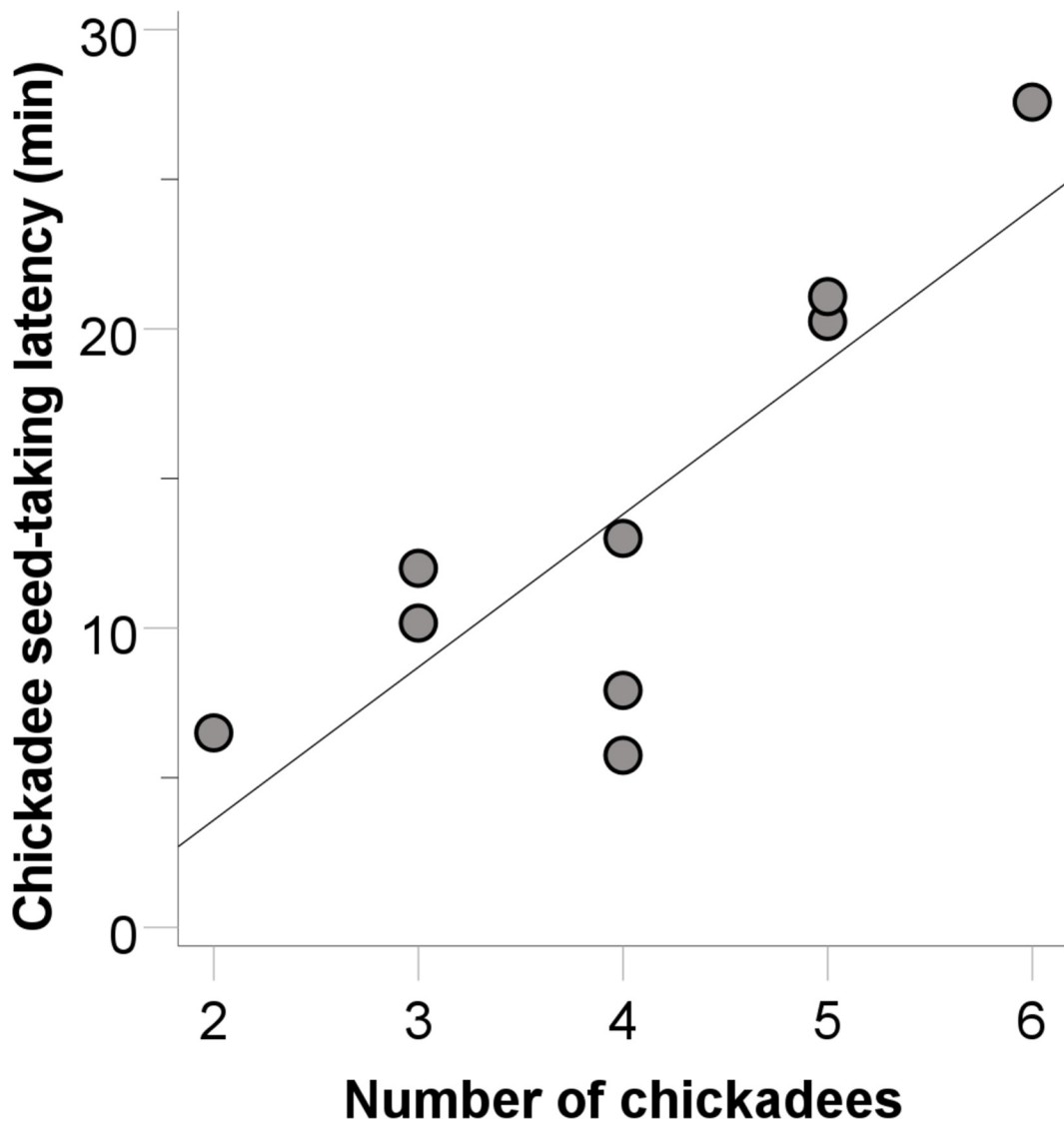

**Fig 4. The relationship between the number of chickadees and the latency to solve the plastic blocks feeder task for successful chickadee flocks (circles).**

(Fig 5A) and a greater number of titmice (Fig 5B) than unsuccessful titmouse flocks. The next largest Wald statistic for titmice was for MSF size (B = -0.268, SE = 0.279, Wald = 0.922, p = 0.337). For successful titmouse flocks, no factor predicted the latency to solve the feeder task (largest t statistic for diversity index: B = -4.219, SE = 6.291, $t_5$ = -0.671, p = 0.513).

### Experiment 3: Separate food source

The mean ± SD (range) number of chickadees in flocks in this experiment was 4.28 ± 1.47 (1–7) individuals. Flocks contained an average of 3.89 ± 1.77 (2–8) titmice and 1.39 ± 0.99 (0–4) nuthatches. Mean MSF size was 11.19 ± 3.03 (5–17) individuals and mean diversity index across flocks was 3.01 ± 0.93 (1.32–4.84). Feeders with the diverse seed mix attracted a median of 4 birds (range 2–7) compared to a median of 3.5 birds (range 2–5) for feeders with the regular seed mix ($n_1$ = 18, $n_2$ = 18, U = 87, p = 0.015).

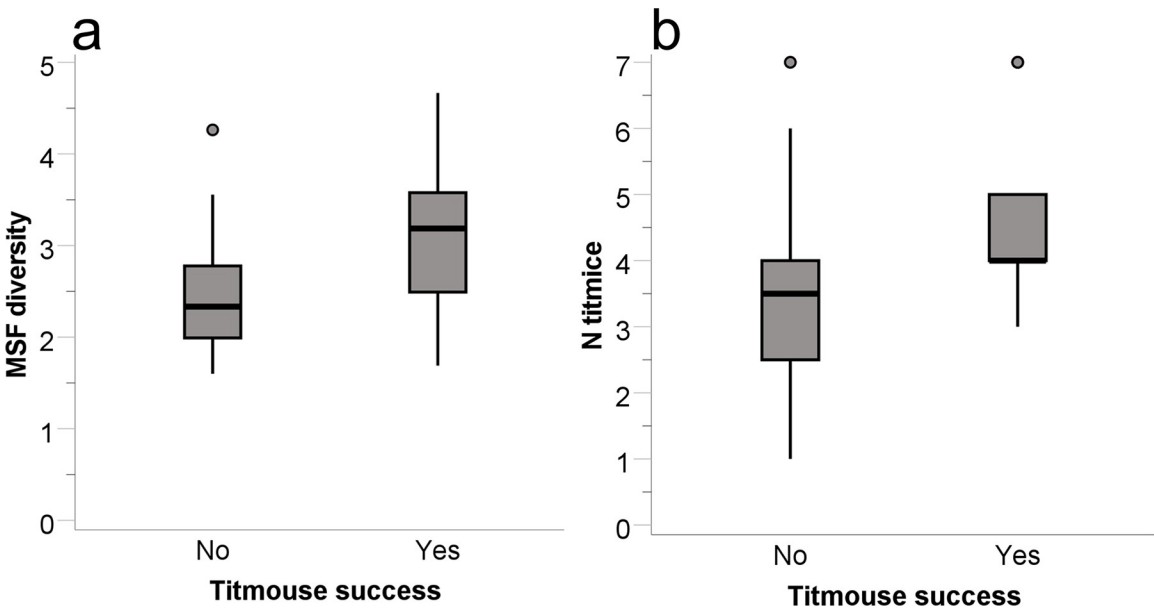

**Fig 5.** Tufted titmouse success at solving the plastic blocks feeder task compared to the diversity index of the flock (a) and to the number of titmice in the flock (b). Data plotted as for Figs 2 and 3.

Carolina chickadee flocks took seed from the separate food source feeder at 16 of the 36 sites. Only MSF size predicted chickadee flock success (B = 1.269, SE = 0.482, Wald = 6.941, p = 0.008, $r^2$ = 0.663). Successful chickadee flocks were from flocks with more total individuals than unsuccessful chickadee flocks (Fig 6A). The next largest Wald statistic for chickadees was

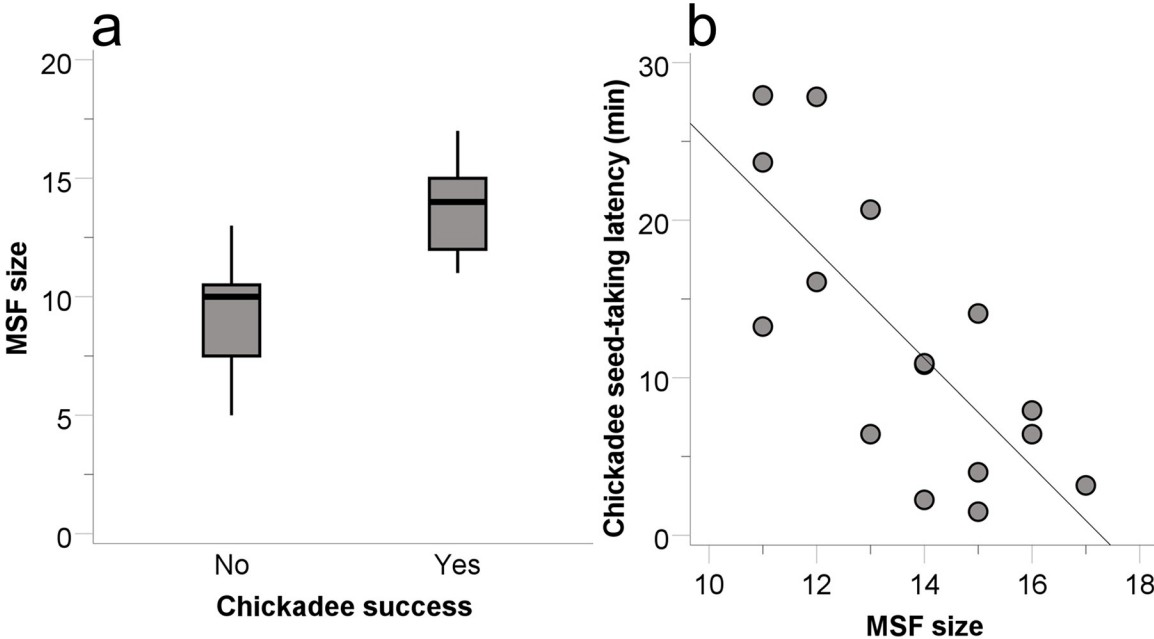

**Fig 6.** Carolina chickadee success at solving the separate food task compared to mixed-species flock (MSF) size (a) and the relationship between the number of chickadees and the latency to solve the separate food task for successful chickadee flocks (b). Data plotted as in Figs 2–4.

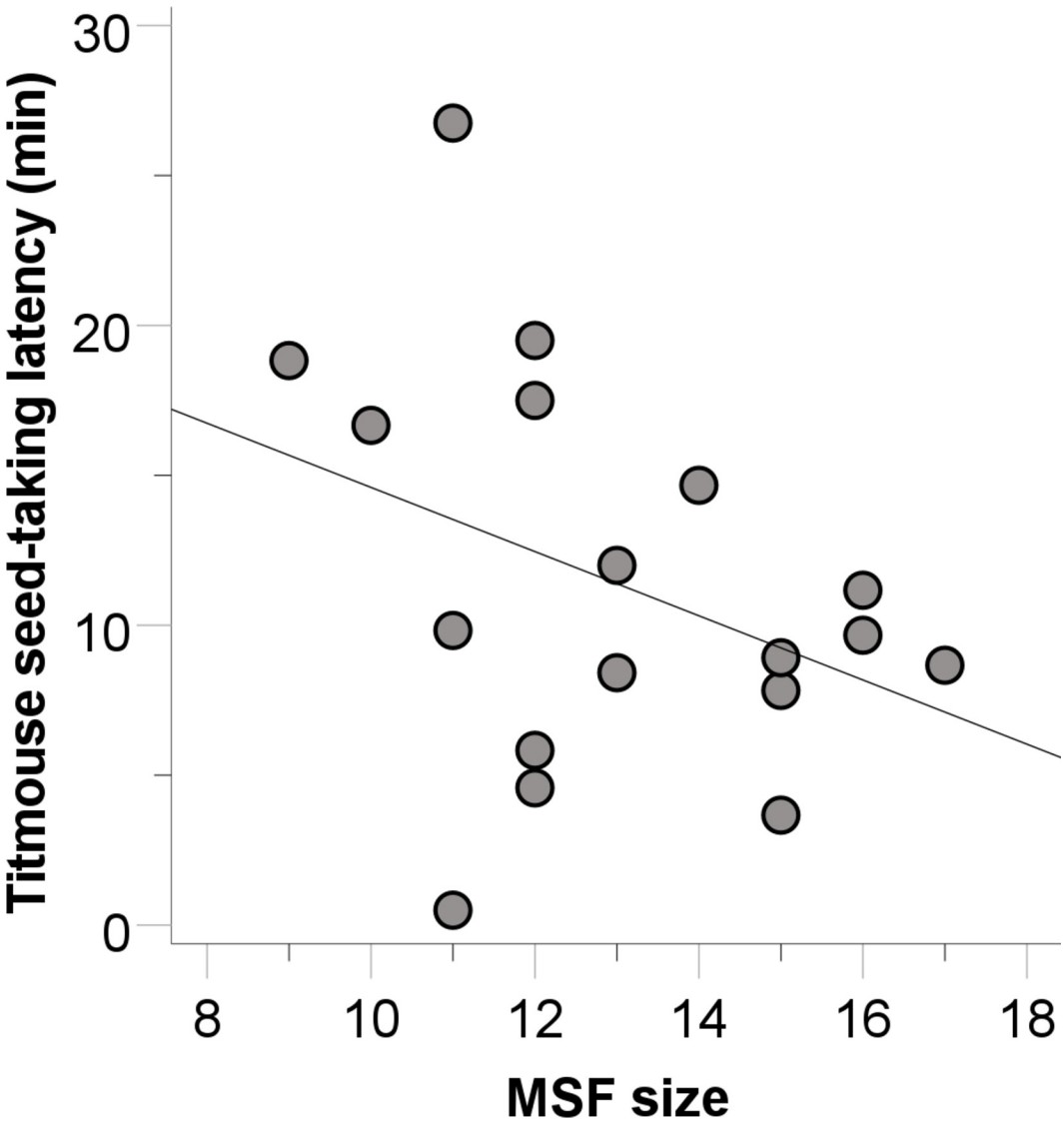

**Fig 7. The relationship between the interspecific flock size and the latency to solve the separate food task for successful titmouse flocks (circles).**

for forest alpha shape (B = -0.001, SE = 0.001, Wald = 1.233, p = 0.267). For successful chickadee flocks, the latency to take seed was also predicted by MSF size (B = -2.446, SE = 1.090, $t_5$ = -2.244, p = 0.049, $r^2$ = 0.560). Chickadees in successful chickadee flocks were quicker to take seed when there were more total birds in their flocks (Fig 6B). The next largest t statistic was for forest alpha shape (B = -0.004, SE = 0.003, $t_5$ = -1.372, p = 0.200).

Tufted titmouse flocks obtained seed from 18 of the 36 separate food source feeders. Titmouse flock success tended to be associated with conspecific flock size (B = 0.871, SE = 0.449, Wald = 3.756, p = 0.053, $r^2$ = 0.510). Successful titmouse flocks tended to be flocks with more titmice than unsuccessful titmouse flocks. For successful titmouse flocks, individuals were quicker to obtain seed in the separate food task if they were in flocks with a greater MSF size (B = -2.890, SE = 1.161, $t_5$ = -2.490, p = 0.028, $r^2$ = 0.138, Fig 7) and a lower mixed-species flock diversity (B = 8.352, SE = 3.784, $t_5$ = 2.207, p = 0.048, $r^2$ = 0.003), though the latter

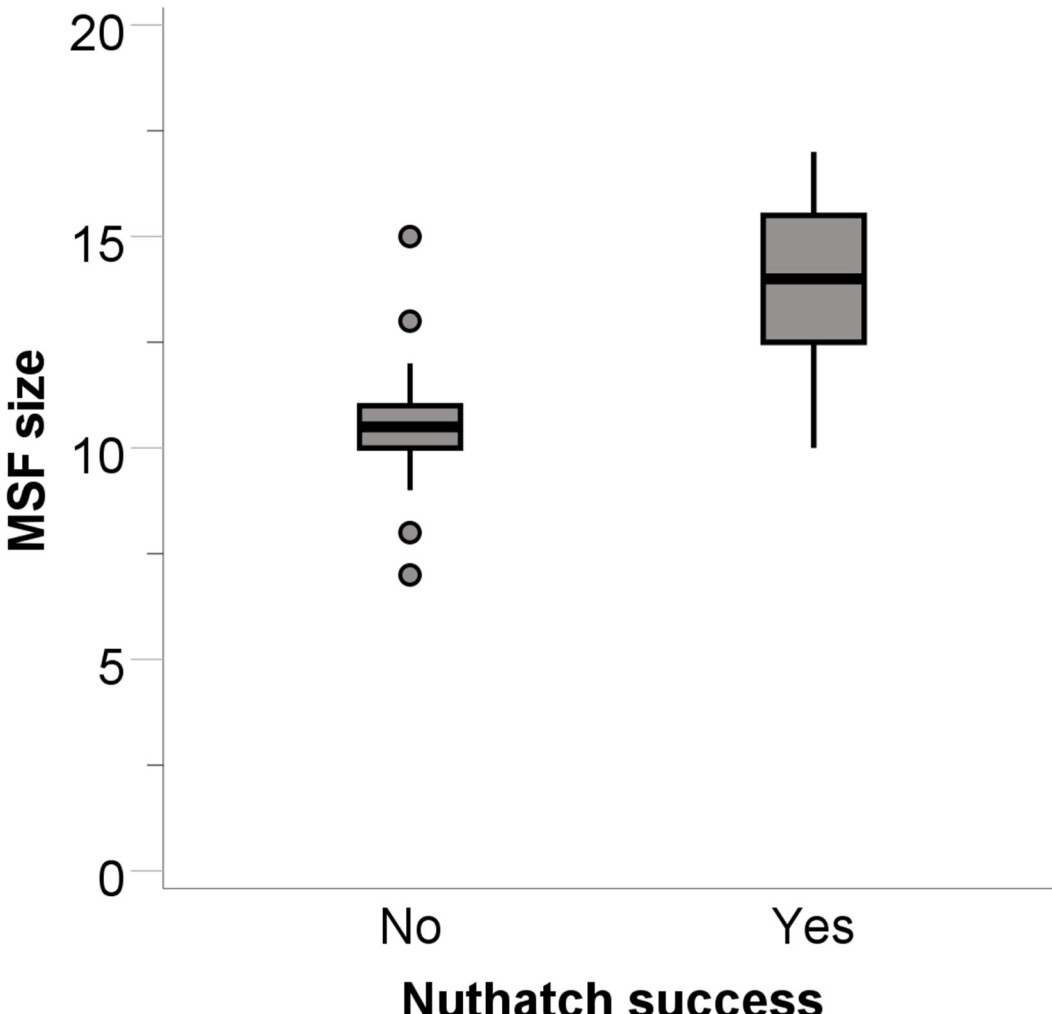

**Fig 8. White-breasted nuthatch success at solving the novel food source task compared to the mixed-species flock (MSF) size.** Data plotted as for Figs 2 and 3.

relationship is marginal at best given the very low effect size. The next largest t statistic for titmice was for number of titmice (B = 1.608, SE = 1.406, $t_5$ = 1.143, p = 0.275).

Of the 26 flocks containing at least one nuthatch, nuthatches took seed at 12 of the novel food source structures. Like for chickadees and titmice in this study, nuthatch success was predicted by MSF size (B = 0.791, SE = 0.361, Wald = 4.796, p = 0.029, $r^2$ = 0.656). Successful nuthatch flocks were in flocks with more total individuals of all species than unsuccessful nuthatch flocks (Fig 8). The next largest Wald statistic for nuthatches was for traffic noise (B = -0.326, SE = 0.229, Wald = 2.030, p = 0.154). No variable predicted latency to obtain a seed from the novel food source for nuthatches from successful nuthatch flocks (largest t statistic for number of nuthatches: B = -6.437, SE = 4.656, $t_5$ = -1.383, p = 0.216).

## Discussion

In all three field experiments using novel feeder tasks, we found that our focal species were sensitive to the size and composition of their mixed-species flocks. Carolina chickadee success at obtaining seed from the novel feeders was associated with greater mixed-species flock

diversity in Experiment 1 (and tended to be in Experiment 2) and with greater total flock size in Experiment 3. Tufted titmouse success at obtaining seed from the novel feeders was associated with the number of conspecifics in Experiment 1 and Experiment 2, and also with the diversity of mixed-species flock composition in Experiment 2. White-breasted nuthatches only solved the novel feeder task in sufficient numbers for analyses in Experiment 3, and success was associated with interspecific flock size. Furthermore, for successful chickadee and titmouse flocks in Experiment 3, individuals exploited seed from the separate food source quicker the larger the interspecific flock size.

These results partially corroborate the findings of our previous study with a novel feeder task [10]. In that earlier study, for both chickadees and titmice, successful flocks had greater mixed-species flock diversity than unsuccessful flocks (nuthatches rarely solved the novel feeder task). Here we found that chickadee success at novel feeder tasks was significantly associated with mixed-species flock diversity in Experiment 1, but with MSF size in Experiment 3. Experiment 3 involved a separate food source placed 3 m away from the birds' normal feeder. Perhaps more individuals and eyes were the key driving factors behind success in the Experiment 3 separate food source design. Intriguingly, for successful chickadees in Experiment 2 involving the plastic blocks structure, latency to obtain seed was shorter the fewer chickadees there were in the flock. It is difficult to explain this effect, except perhaps as being related to increased dominant-subordinate interactions disrupting foraging in larger flocks (though we have not noticed this being an issue in other studies). For titmice, as with the earlier study [10], greater mixed-species flock diversity was associated with success for the plastic blocks feeder task of Experiment 2.

Taken together, our results provide partial support for H1, the pool of competence benefit hypothesis, and for H2, the group size benefit hypothesis (for both conspecific group size in titmice and for MSF size in all three species). We found no support for H3, the habitat density benefit hypothesis, or H4, the traffic noise distraction hypothesis. This contrasts with a recent pair of experiments at these same feeders that involved a context of risk: one experiment played back a string of alarm calls of tufted titmice near the feeders and the other experiment presented a model of a screech owl directly on the feeders [18]. In that study, we found that vegetation density predicted return-to-foraging behavior after the playback in chickadees and nuthatches; for both species, latency to resume foraging was longer the denser the vegetation. Additionally, there was a tendency for the duration of mobs to the screech owl model to be shorter at feeders with higher levels of traffic noise. We have long known that physical environmental factors like nearby cover (dense habitat close to the individual) affect rates of vigilance and how individuals respond to predator threat [30,31]. Why these physical environmental factors seem to be more important to risk- and predator-related contexts than to novel feeder tasks awaits future testing. It is additionally intriguing that mixed-species flock diversity seemed to have minimal influence on those risk- and predator-related contexts [18], but clear influence on these novel feeder tasks (this study and [10]).

We have known for decades from both observational and experimental studies that human groups regularly benefit when they are more diverse (healthcare decisions: [32]; marketplace and business decisions: [33–36]; online information gathering: [37]). These benefits are thought to stem from cognitive diversity among group members [38,39]. Models reveal, furthermore, that if groups are not cognitively diverse but are demographically diverse (e.g., gender, ethnicity, age, culture, etc.), those groups can still benefit in comparison to more homogeneous groups due to differences in in-group dependence and conformity [40,41]; see also [42]. Taken together, data from human studies suggest general benefits of diversity in groups to 'bottom line' interests of those groups, though depending on the specific contexts those benefits can be small [43].

Data from human studies may inform future work on diversity in groups of non-human animals, including mixed-species groups. For example, humans living in more diverse communities exhibit greater levels of pro-social behavior [44]. It has long been known that levels of competition among members of conspecific-only groups are typically higher than for members of similarly-sized mixed-species groups, likely due to the much broader foraging and resource niches of the different species [3]. One barrier to human groups achieving benefits of diversity occurs when there is ineffective communication among group members [39]. In mixed-species groups, as long as group members are able to communicate effectively with one another, individuals in those groups should benefit from within-group diversity [45]. For example, red-breasted nuthatches, *Sitta canadensis*, respond adaptively to variation in calling behavior of black-capped chickadees, *Poecile atricapillus* [46] and northern cardinals, *Cardinalis cardinalis*, show similar adaptive anti-predator responses to playbacks of different calls of tufted titmice that vary in level of associated risk due to predation [47]. We increasingly understand how signals and cues used by senders and receivers across species boundaries facilitate and maintain mixed-species groups [48–50]. A recent study, for example, has found that young crested tits, *Lophophanes cristatus*, are more likely to produce alarm calls in the winter and early spring months when they are in flocks with willow tits, *Poecile montanus*, than when they are alone or in flocks with just conspecifics [51].

It perhaps should not be surprising that the three focal species of our three experiments here were so sensitive to the size and composition of their mixed-species flocks. After all, these birds spend a great deal of their daily time budgets–particularly in the overwintering months when our studies were conducted–in these flocks [52,53]. Greater research on how these mixed-species flocks–and other mixed-species groups–move while remaining cohesive should prove essential to understanding how they effectively avoid predation and find food [26,48,54]. Given that tufted titmice, white-breasted nuthatches, and black-capped chickadees were among the best problem solvers in a recent comparative study [12], these seem like important species to which individuals in these mixed-species flocks should attend.

Mixed-species groups can play a major role in the broader community structure and functioning–the presence of a core set of species in an area often impacts the presence and behavior of other species in that area [4]. Increased species richness in groups can lead to greater robustness of food webs in the face of drastic environmental changes [55]. Variation in habitat characteristics can also influence the ways in which species in mixed-species groups interact with one another [56]. Thus, although we found minimal influence of physical environmental factors on chickadee, titmouse, and nuthatch success in our three novel feeder studies, we believe future work is warranted to assess the influence of vegetation density and background noise on mixed-species group structure and functioning. This need grows in importance as habitats are increasingly disturbed and threatened by human activities.

## Supporting information

**S1 Fig. We recorded background noise levels 4–6 times at each feeder between November 2021 and March 2022.** When recording noise levels at each feeder, we noted the noise level reading on the SPL meter (see text) at 20, 30, 40, 50, and 60 sec during a one minute period when there were no airplanes overhead and only on calm mornings. The median noise level reading was coded for that day at that feeder. Data here are plotted as means (circles) and SDs (whiskers) of those daily median noise levels for each feeder.
(TIF)

**S1 File. SOM1: Spreadsheet of data from Experiment 1 –transparent tube feeder study.**
(XLSX)

**S2 File. SOM2: Spreadsheet of data from Experiment 2 –plastic blocks feeder study.**
(XLSX)

**S3 File. SOM3: Spreadsheet of data from Experiment 3 –separate food source study.**
(XLSX)

## Acknowledgments

The authors thank the staff at the University of Tennessee Forest Resources, Research & Education Center for permitting us to carry out our study on their grounds. Thanks to Alexandria Hardy for help with data collection in Study 2, to Árpád Nyári for help with lidar data collection, and to Rami Abu-Shehadeh and Kelsie Childress for help with coding video files for inter-observer reliability measures. We thank Scott Benson, Heather Brooks, Eric Frazier, S Ryan Risner, Zaharia Selman, and especially Kathryn Sieving for helpful critiques of earlier drafts of this manuscript.

## Author Contributions

**Conceptualization:** Todd M. Freeberg.

**Data curation:** Todd M. Freeberg, Colton B. Adams, Charles A. Price, Monica Papeş.

**Formal analysis:** Todd M. Freeberg, Colton B. Adams, Charles A. Price, Monica Papeş.

**Funding acquisition:** Todd M. Freeberg.

**Investigation:** Todd M. Freeberg, Colton B. Adams, Charles A. Price, Monica Papeş.

**Methodology:** Todd M. Freeberg, Colton B. Adams, Charles A. Price, Monica Papeş.

**Project administration:** Todd M. Freeberg.

**Resources:** Todd M. Freeberg.

**Software:** Todd M. Freeberg, Colton B. Adams, Charles A. Price, Monica Papeş.

**Supervision:** Todd M. Freeberg.

**Validation:** Todd M. Freeberg.

**Visualization:** Todd M. Freeberg.

**Writing – original draft:** Todd M. Freeberg.

**Writing – review & editing:** Todd M. Freeberg, Colton B. Adams, Charles A. Price, Monica Papeş.

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
