## [Decision Letter · Decision Letter 0]

31 Jan 2024

PONE-D-23-39771Mixed-species flock sizes and compositions influence flock members’ success in three field experiments with novel feedersPLOS ONE

Dear Dr. Freeberg,

Thank you for submitting your manuscript to PLOS ONE. After careful consideration, we feel that it has merit but does not fully meet PLOS ONE’s publication criteria as it currently stands. Therefore, we invite you to submit a revised version of the manuscript that addresses the points raised during the review process. You should edit your article according to the opinions of the referees and upload it to the system.

We look forward to receiving your revised manuscript.

Kind regards,

Esin Ebru Onbaşılar

Academic Editor

PLOS ONE

Journal Requirements:

"Funding that supported this work was obtained from the Department of Psychology and the College of Arts & Sciences of the University of Tennessee, Knoxville."

5. We note that you have referenced (Freeberg, unpublished data) on page 6,  which has currently not yet been accepted for publication. Please remove this from your References and amend this to state in the body of your manuscript: (ie “Bewick et al. [Unpublished]”) as detailed online in our guide for authors

Reviewers' comments:

Reviewer's Responses to Questions

**Comments to the Author**

1. Is the manuscript technically sound, and do the data support the conclusions?

Reviewer #1: Yes

2. Has the statistical analysis been performed appropriately and rigorously? 

Reviewer #1: Yes

3. Have the authors made all data underlying the findings in their manuscript fully available?

Reviewer #1: Yes

4. Is the manuscript presented in an intelligible fashion and written in standard English?

Reviewer #1: Yes

5. Review Comments to the Author

Reviewer #1: This is a very tightly written article on a rigorously designed set of experiments that advances and improves upon, and is consistent with, earlier work. A broad audience will appreciate this paper because of its implications for understanding how social group structure/diversity influences problem solving by higher vertebrates (including humans). I only have a few suggestions for improvement.

(1) at line 54 (or before) it would be valuable to cite a new study of bird problem-solving by J-N Audet et al. https://doi.org/10.1126/science.adh3428 - this paper shows that TUTI, WBNU and the close kin of CACH (BCCH) rank at the top of all birds in terms of problem-solving ability. It particularly underscores hypothesis 1 and 2 and helps explain why TUTI rely mostly on other tuti (as they rank highest of all).

(2) Line 92 - a better study than crabs to cite here would be Grade and Sieving 2016 (https://doi.org/10.1098/rsbl.2016.0113) or other similar/recent bird studies.

(3) Lines 436-440 - This portion of this paragraph would benefit from addition of well-known insights concerning animals reliance on cover when in the presence of explicit predation risks. The point being made is that it seems surprising that in this feeding study birds were not sensitive to the availability of cover when they were in a study that imposed explicit predation risks at feeders. But their results do not surprise me at all - when animals perceive an explicit threat, cover becomes extremely important if it is available (Steve Lima's works) and overall risk-taking declines (lots of papers - this one should lead you to the best sources to discuss via seeing who cites it more recently (Huang et al. 2011; https://doi.org/10.1093/beheco/arr212). Given the absence of explicit risks here, these birds' familiarity with their local (habitual) feeder sites would soften their caution in approaching when they don't detect a threat - neither noises nor vegetation configurations they are used to on a daily basis should influence them overly much when they are in groups. Basically, it is not a surprising result, but rather explicable.

(4) Line 464 suggests we need to know more about cross-species information sharing - but there is a very large literature on it these days (heterospecific social information). Surely this could be less of a question and more of a conclusion concerning the results making sense - given the high importance of heterospecific social information in the lives of both flocking and non-flocking birds.

Overall - a very cool study with implications for human social behavior and problem solving in diverse groups.

6. PLOS authors have the option to publish the peer review history of their article (what does this mean?). If published, this will include your full peer review and any attached files.

Reviewer #1: **Yes: **Kathryn Sieving

---

## [Author Response · Author response to Decision Letter 0]

7 Mar 2024

Dear Dr. Onbaşılar:

We thank you and the reviewers for helpful comments on our original submission. We have revised the manuscript based upon those comments and questions raised. Below is the entire text of your decision letter related to a revised manuscript. Following each point, we outline changes we made in the text to address the concern and indicate where those changes can be found by line number in the clean copy uploaded. If we can provide additional information, please let me know!

Journal Requirements:

Done!

We will have our data tables for the three experiments available at the PLoS ONE website for our study as supplemental online material.

"Funding that supported this work was obtained from the Department of Psychology and the College of Arts & Sciences of the University of Tennessee, Knoxville."

Done.

Done. 

5. We note that you have referenced (Freeberg, unpublished data) on page 6, which has currently not yet been accepted for publication. Please remove this from your References and amend this to state in the body of your manuscript: (ie “Bewick et al. [Unpublished]”) as detailed online in our guide for authors

We deleted this problematic sentence since it was not really needed and since we ended up finding that the diverse seed mix did in fact attract more total species to the feeders in these experiments (data reported in original results sections).

Done.

Done.

Reviewers' comments:

Reviewer's Responses to Questions

Comments to the Author

1. Is the manuscript technically sound, and do the data support the conclusions?

Reviewer #1: Yes

2. Has the statistical analysis been performed appropriately and rigorously?

Reviewer #1: Yes

3. Have the authors made all data underlying the findings in their manuscript fully available?

Reviewer #1: Yes

4. Is the manuscript presented in an intelligible fashion and written in standard English?

Reviewer #1: Yes

5. Review Comments to the Author

Reviewer #1: This is a very tightly written article on a rigorously designed set of experiments that advances and improves upon, and is consistent with, earlier work. A broad audience will appreciate this paper because of its implications for understanding how social group structure/diversity influences problem solving by higher vertebrates (including humans). I only have a few suggestions for improvement.

(1) at line 54 (or before) it would be valuable to cite a new study of bird problem-solving by J-N Audet et al. https://doi.org/10.1126/science.adh3428 - this paper shows that TUTI, WBNU and the close kin of CACH (BCCH) rank at the top of all birds in terms of problem-solving ability. It particularly underscores hypothesis 1 and 2 and helps explain why TUTI rely mostly on other tuti (as they rank highest of all).

Thank you for the helpful suggestion – text added and reference cited in lines 56-60 and also in the discussion section in lines 486-489.

(2) Line 92 - a better study than crabs to cite here would be Grade and Sieving 2016 (https://doi.org/10.1098/rsbl.2016.0113) or other similar/recent bird studies.

We kept the hermit crab study as it involved acoustic noise but a visual stimulus and so could not be masking. We did add the Grade & Sieving (2016) study here – should have done that in the first place! Change in lines 98-99.

(3) Lines 436-440 - This portion of this paragraph would benefit from addition of well-known insights concerning animals reliance on cover when in the presence of explicit predation risks. The point being made is that it seems surprising that in this feeding study birds were not sensitive to the availability of cover when they were in a study that imposed explicit predation risks at feeders. But their results do not surprise me at all - when animals perceive an explicit threat, cover becomes extremely important if it is available (Steve Lima's works) and overall risk-taking declines (lots of papers - this one should lead you to the best sources to discuss via seeing who cites it more recently (Huang et al. 2011; https://doi.org/10.1093/beheco/arr212). Given the absence of explicit risks here, these birds' familiarity with their local (habitual) feeder sites would soften their caution in approaching when they don't detect a threat - neither noises nor vegetation configurations they are used to on a daily basis should influence them overly much when they are in groups. Basically, it is not a surprising result, but rather explicable.

We added text in lines 442-444 to address this important point and added two classic Lima references here.

(4) Line 464 suggests we need to know more about cross-species information sharing - but there is a very large literature on it these days (heterospecific social information). Surely this could be less of a question and more of a conclusion concerning the results making sense - given the high importance of heterospecific social information in the lives of both flocking and non-flocking birds.

Wording changed in this section to minimize the ‘question’ framing in the original manuscript – see lines 471-479.

Overall - a very cool study with implications for human social behavior and problem solving in diverse groups.

6. PLOS authors have the option to publish the peer review history of their article (what does this mean?). If published, this will include your full peer review and any attached files.

Do you want your identity to be public for this peer review? For information about this choice, including consent withdrawal, please see our Privacy Policy.

Reviewer #1: Yes: Kathryn Sieving

---

## [Editor Report · Decision Letter 1]

14 Mar 2024

Mixed-species flock sizes and compositions influence flock members’ success in three field experiments with novel feeders

PONE-D-23-39771R1

Dear Dr. Freeberg,

We’re pleased to inform you that your manuscript has been judged scientifically suitable for publication and will be formally accepted for publication once it meets all outstanding technical requirements.

Kind regards,

Esin Ebru Onbaşılar

Academic Editor

PLOS ONE
---

## [Editor Report · Acceptance letter]

24 Mar 2024

PONE-D-23-39771R1 

PLOS ONE

Dear Dr. Freeberg, 

I'm pleased to inform you that your manuscript has been deemed suitable for publication in PLOS ONE. Congratulations! Your manuscript is now being handed over to our production team.

Kind regards, 

on behalf of

Dr. Esin Ebru Onbaşılar 

Academic Editor

PLOS ONE